# AG-ViT: Atlas-Guided Vision Transformer for Predicting Cognitive Decline*

## Abstract

The use of resting state fMRI (rs-fMRI) to improve the diagnosis and treatment of neurodegenerative diseases has increased dramatically in recent years. Despite evident progress, producing accurate predictions from rs-fMRI scans remains challenging due to the data's high dimensionality and the limited number of samples. In this work, our aim is to estimate the probability of cognitive decline within a given time frame based on rs-fMRI scans of Alzheimer's patients. Accurate predictions of disease trajectory can guide medical decision-making and contribute to personalized medicine. To this end, we design a vision transformer to obtain low-dimensional embeddings of rs-fMRI scans. These embeddings are used to train a network that estimates the probability of cognitive decline. By testing our approach on scans from the Alzheimer's Disease Neuroimaging Initiative, we show that models trained on our transformer-based features improve F1-score by 9–15 percentage points compared to those trained on handcrafted connectivity features. For interpretability, we develop a simple yet effective method to identify brain regions whose fMRI-derived signal significantly impacted model predictions. The results identified a set of brain regions, some recognized for their early involvement in AD and others for their relative resilience to AD pathology.

## 1 Introduction

In recent years, analysis of resting state fMRI (rs-fMRI) scans has become a key tool for studying neurodegenerative disorders such as Alzheimer's disease, dementia, and Parkinson's disease (Filippi & Filippi, 2016; Filippi et al., 2019). Many longitudinal studies have utilized rs-fMRI scans to investigate the degradation of connectivity patterns between brain regions throughout disease progression (Alorf & Khan, 2022; Hohenfeld et al., 2018). These investigations have provided important insights into the pathophysiology of neurodegenerative diseases and the role of established neuronal networks in their trajectory.

While significant progress has been achieved, computational approaches for rs-fMRI analysis still face substantial limitations. fMRI scans measure Blood-Oxygen-Level-Dependent (BOLD) signals, which serve as a surrogate for brain activity. Each scan consists of $\sim 10^5$ voxels, each with several hundred time samples of the BOLD signal. An rs-fMRI dataset, however, often contains fewer than $10^3$ scans. Analyzing such limited data is further complicated by the considerable variability in brain activity, even among patients with comparable clinical characteristics.

To address the variability of brain activity, Finn et al. (2015) demonstrated that functional connectivity (FC) patterns between brain regions are more stable than raw brain activity and can serve as a unique *fingerprint* for predicting phenotypes such as fluid intelligence. FC between regions is typically estimated through the correlation of their corresponding fMRI time series Wang et al. (2014). Relying solely on pairwise FC may be suboptimal, however, in cases where alterations involve coordinated changes across multiple brain regions within a neuronal network. Detecting such networks, particularly those associated with the progression of neurodegenerative disorders, remains a major challenge in fMRI analysis.

---

*Code will be publicly available upon acceptance.

In this work, our main task is to estimate the probability of cognitive decline within a given timeframe for patients diagnosed with Alzheimer's disease. Our dataset consists of approximately 900 rs-fMRI scans obtained from the Alzheimer's Disease Neuroimaging Initiative (ADNI) repository. With the recent approval of new treatments, accurate predictions of disease progression can guide personalized treatment regimens and optimize clinical strategies. Specifically, we make the following contributions:

- We propose a self-supervised attention-based pretraining strategy to learn low-dimensional representations of rs-fMRI scans. To reduce the number of parameters in our architecture, we design an atlas-based decoder, training the network to predict the atlas representation of each scan.

- We train a neural network to predict cognitive decline from the learned embeddings. Our approach achieves substantially higher accuracy compared to similar models trained directly on FC matrices, demonstrating that transformer-based features are more informative than handcrafted FC features.

- We show that the accuracy of the network can be further improved by employing a test-time adaptation mechanism.

- We introduce a simple yet effective approach for interpreting our transformer model, identifying brain regions that significantly influence predictions. We compare the pattern of brain regions that significantly contribute to model performance with the characteristic pattern of amyloid $\beta$ ($A\beta$) accumulation in Alzheimer's disease, drawing insights into the biological relevance of the model.

## 2 RELATED WORK

As mentioned, a common strategy to mitigate the variability in brain activity between samples is to engineer features by computing pairwise functional connectivity (FC) matrices, which measure the statistical dependence between time-series of atlas-defined brain regions (Finn et al., 2015; Rosenberg et al., 2016; Friston, 2011). Building upon these engineered features, a variety of deep learning models were developed for different tasks in neuroimaging analysis. Architectures including Convolutional Neural Networks (CNNs), Graph Neural Networks (GNNs), and attention-based Deep Neural Networks were designed to learn patterns from engineered representations like FC matrices or ICA-based spatial maps. These models were used for tasks such as disease classification and prediction of clinical symptom progression (Lin et al., 2022; Sarraf et al., 2016; Sheynin et al., 2021; Parisot et al., 2019; Kawahara et al., 2017). Recent approaches have also applied Transformers to measurements from specific regions of interest (Kan et al., 2022).

To move beyond handcrafted features, recent efforts have focused on end-to-end deep learning models, particularly Transformers, which excel at modeling the long-range dependencies characteristic of brain networks. A prominent direction is self-supervised pretraining on large, unlabeled fMRI datasets. Several models learn representations by reconstructing raw voxel information, such as the framework proposed by Malkiel et al. (2022) and the Swin 4D fMRI Transformer (SwiFT) (Kim et al., 2023). Another powerful approach is masked signal modeling, exemplified by BrainLM, a foundation model pretrained on thousands of hours of fMRI data by predicting activity in masked brain regions (Ortega et al., 2024). Our work takes this pretraining trajectory in a new direction by introducing a novel, atlas-guided objective. We hypothesize that reconstructing functionally meaningful regional dynamics, rather than relying on raw voxel-wise or masked regional signals, offers a more robust and data-efficient pretraining task. This approach leverages the well-validated Schaefer cortical parcellation, which segments the brain into parcels associated with networks characterized by strong intra-regional functional connectivity.

## 3 METHOD

We describe our approach for predicting cognitive decline from rs-fMRI data. In Section 3.1, we formulate the problem and outline our primary and secondary objectives. Section 3.2

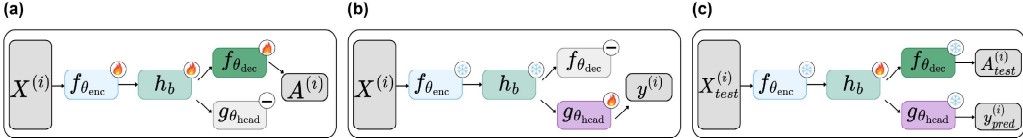

Figure 1: **Training stages of the AG-ViT framework. (a) Stage 1: Self-supervised pretraining** learns to reconstruct atlas representations $\mathbf{A}^{(i)}$ from 4D fMRI scans $\mathbf{X}^{(i)}$ through encoder $f_{\theta_{\text{enc}}}$ and decoder $f_{\theta_{\text{dec}}}$, creating compact bottleneck features $\mathbf{h}_b$. The task head $g_{\theta_{\text{head}}}$ is not used . **(b) Stage 2: Supervised fine-tuning** freezes the pretrained encoder and trains task-specific heads $g_{\theta_{\text{head}}}$ for downstream predictions $y^{(i)}$. The decoder is not used. **(c) Test-time adaptation** iteratively refines the bottleneck representation by minimizing atlas reconstruction error using the frozen decoder, improving final predictions $\hat{y}^{(i)}_{\text{pred}}$. Flame icons indicate trainable parameters; snowflakes indicate frozen parameters; minus icons indicate unused components.

introduces the overall two-stage framework, including our novel self-supervised pretraining strategy and supervised fine-tuning. We then describe the architectural components of our 4D Vision Transformer in Section 3.3 and the test-time adaptation mechanism in Section 3.4. Finally, our main experimental results on the cognitive decline task are presented in Section 4. Results for predicting sex and age are provided in the supplementary material.

## 3.1 Problem Formulation and Objectives

We utilize data from the Alzheimer's Disease Neuroimaging Initiative (ADNI) database (Jack Jr et al., 2008; Mueller et al., 2005), a longitudinal multicenter study designed to develop clinical, imaging, genetic, and biochemical biomarkers for the early detection and tracking of Alzheimer's disease. Let $\mathcal{D} = \{(\mathbf{X}^{(i)}, y^{(i)})\}_{i=1}^{N}$ denote our dataset, where $\mathbf{X}^{(i)} \in \mathbb{R}^{T \times H \times W \times D}$ represents a rs-fMRI scan of patient $i$, with $T$ temporal samples and $(H, W, D)$ spatial dimensions, and $y^{(i)}$ is our task's label.

**Primary Objective:** This work addresses the key challenge of predicting Alzheimer's disease progression within a specified time frame. We assess progression using the Clinical Dementia Rating (CDR) score, a widely adopted cognitive measure for Alzheimer's disease (Hughes et al., 1982; Morris, 1993). In ADNI, each patient typically undergoes a CDR assessment once per year. Our objective is to predict whether a patient's CDR score will change within a given time period.

**Secondary Objective:** Beyond prediction accuracy, we aim to identify brain regions that significantly affect the outcome of our prediction.

## 3.2 Overall Framework

To meet our objectives, we develop a transformer-based pipeline that is trained in two stages:

**Stage 1: Self-supervised pretraining via atlas reconstruction.** The input for this stage consists of the original 4D fMRI scans $\{X^{(i)}\}$. The self-supervised learning target is a 2D atlas representation, denoted $A^{(i)} \in \mathbb{R}^{R \times T}$. The atlas time series are computed using the Schaefer 2018 parcellation (Schaefer et al., 2018), which divides the cerebral cortex into $R = 200$ functionally coherent regions based on resting-state connectivity patterns. This parcellation scheme has been validated to align with known functional networks and provides a biologically meaningful dimensionality reduction from voxel space (∼100K voxels) to regional space ($R = 200$ regions). Each region's activation time series is computed by spatially averaging the signals of all voxels within that region's corresponding spatial mask. The result is an $R \times T$ matrix, where each row contains a single region's temporal dynamics and each column represents the brain-wide activation pattern at a specific time point.

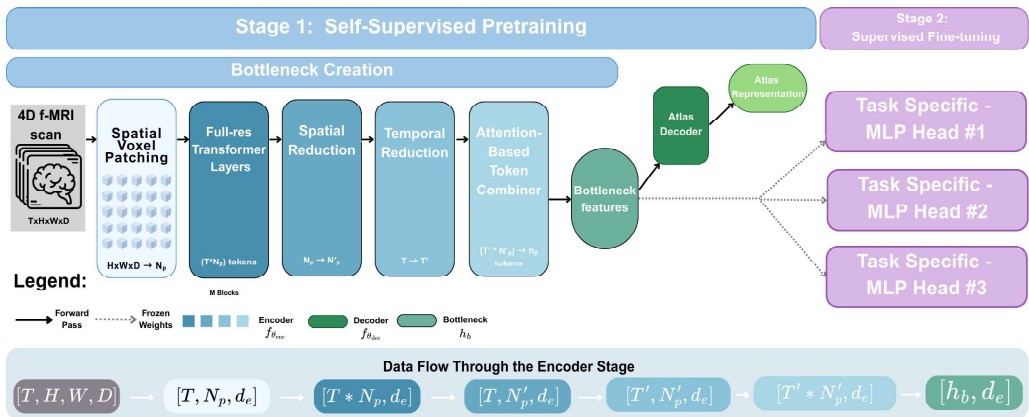

Figure 2: **Architectural overview of the AG-ViT framework.** Our model is trained in two stages. **Stage 1: Self-Supervised Pretraining.** An encoder-decoder architecture with progressive spatiotemporal reduction maps a 4D fMRI scan to a compact set of bottleneck features. The decoder's objective is to reconstruct an atlas representation of each scan. The bottom panel illustrates the progressive reduction of data dimensionality through the encoder. **Stage 2: Supervised Fine-tuning.** The pretrained encoder is frozen, acting as a universal feature extractor. The learned bottleneck representations are then used to train task-specific MLP heads for various downstream predictions.

We learn an encoder function $f_{\theta_{\text{enc}}} : \mathbb{R}^{T \times H \times W \times D} \to \mathbb{R}^{d_b}$ that maps the high-dimensional fMRI volume to a compact bottleneck representation $\mathbf{h}_b$. This encoder is trained jointly with a decoder $f_{\theta_{\text{dec}}} : \mathbb{R}^{d_b} \to \mathbb{R}^{R \times T}$ to reconstruct, for each scan $X^{(i)}$, its corresponding atlas representation $\mathbf{A}^{(i)}$. The parameters for both the encoder and decoder are optimized on unlabeled fMRI data by minimizing the reconstruction error:

$$\theta_{\text{enc}}^*, \theta_{\text{dec}}^* = \arg\min_\theta \mathbb{E}_{\mathbf{X} \sim \mathcal{D}} \| f_{\theta_{\text{dec}}}(f_{\theta_{\text{enc}}}(\mathbf{X})) - \mathbf{A} \|^2, \tag{1}$$

where $\| \cdot \|$ denotes the Frobenius norm.

**Stage 2: Supervised training of prediction head.** After pretraining, we freeze the parameters of the encoder $f_{\theta_{\text{enc}}^*}$ and use it to map each input sample $\mathbf{X}^{(i)}$ to its low-dimensional bottleneck representation $\mathbf{h}_b^{(i)} = f_{\theta_{\text{enc}}^*}(\mathbf{X}^{(i)})$. A task-specific prediction head $g_{\theta_{\text{head}}}$ is then trained on the labeled data to predict the labels $y^{(i)}$ from the low-dimensional embeddings $\mathbf{h}_b^{(i)}$. The parameters $\theta_{\text{head}}$ are optimized to minimize the cross-entropy loss function, denoted $\mathcal{L}_{\text{task}}$.

$$\hat{\theta}_{\text{head}} = \arg\min_{\theta_{\text{head}}} \mathcal{L}_{\text{task}}(g_{\theta_{\text{head}}}(f_{\theta_{\text{enc}}^*}(\mathbf{X})), y). \tag{2}$$

Figure 1 provides an illustration of the two-step procedure, and the subsequent Test-Time adaptation mechanism described in Section 3.4.

### 3.3 ARCHITECTURE

Our model is a 4D Vision Transformer (Vaswani et al., 2017; Dosovitskiy et al., 2020) with a progressive reduction encoder. As illustrated in Figure 1 and described in detail in Section 3.2, the framework operates in two main stages: self-supervised pretraining via atlas reconstruction, followed by supervised fine-tuning with task-specific heads.

**Encoder Architecture.** The encoder is designed to efficiently map the high-dimensional 4D fMRI input ($\mathbf{X} \in \mathbb{R}^{T \times H \times W \times D}$) to a compact bottleneck representation ($\mathbf{h}_b$). Its key components are as follows:

- **Spatiotemporal Patch Embedding.** For each timepoint, the scan is partitioned into non-overlapping 3D spatial patches of size $6 \times 6 \times 6$. Each patch is flattened and linearly

Figure 3: **Test-Time Adaptation (TTA) Process.** The initial bottleneck, $\mathbf{h}_0$, is iteratively refined over $K$ steps. In each step, the decoder reconstructs the atlas, and the bottleneck is updated to minimize the reconstruction error against the subject's true atlas. The final adapted bottleneck, $\mathbf{h}_K$, is then used for task prediction.

projected into a token embedding. Following common practice, to preserve the 4D structure, we augment these tokens with separate $2D$ sinusoidal positional encoding for their spatial and temporal coordinates.

- **Progressive Reduction.** The sequence of tokens is processed through four stages that progressively reduce dimensionality. The initial stage consists of four standard Transformer blocks operating at the full spatiotemporal resolution. The subsequent stages systematically compress the representation. First, a learned spatial reduction module reduces the number of spatial patches per time point from $N_p$ to a smaller set $N_p'$. This is followed by a similar temporal reduction module that compresses the temporal dimension from $T$ to $T'$, see illustration in Fig. 6(A).

- **Attention-Based Bottleneck.** In the final stage of the encoder, we employ an attention-based aggregation mechanism inspired by PatchMerger (Yu et al., 2022). As shown in Figure 2b, a small set of learned query tokens attends to the spatiotemporal feature map, aggregating the information into a compact, fixed-size vector. This mechanism inherently captures the statistical dependencies and relationships between different spatial-temporal patches of brain activity. The attention weights explicitly quantify the relevance or influence of each brain region's activity on the learned query tokens, thereby directly reflecting the functional connectivity patterns between them (Wei et al., 2024; Kim et al., 2024).

**Decoder.** We train a lightweight multi-layer perceptron (MLP) to reconstruct the atlas $A^{(i)}$ from the corresponding bottleneck representation $h_b^{(i)}$ by minimizing the loss defined in equation 1.

**Task Heads.** During supervised training, the pretrained encoder is frozen, and shallow 2-layer MLP heads are trained on the bottleneck features for each downstream classification task.

### 3.4 TEST-TIME ADAPTATION

Inspired by the success of test-time adaptation techniques in computer vision (Gandelsman et al., 2022; Sun et al., 2020), we apply this principle to fMRI analysis and find that it yields significant improvements by leveraging subject-specific nuances. We use the subject's own atlas-based activations as a self-supervised signal to iteratively refine the bottleneck representation. The process is illustrated in Figure 3. Using the frozen pretrained decoder, we iteratively refine the bottleneck representation starting from the initial $h^0 = f_{\theta_{enc}^*}(X_{\text{test}})$. The bottleneck is optimized via gradient descent for $K$ steps to better reconstruct the test subject's atlas:

$$\mathbf{h}^{k+1} = \mathbf{h}^k - \eta \nabla_{\mathbf{h}} \| f_{\theta_{\text{dec}}^*}(\mathbf{h}^k) - \mathbf{A}_{\text{test}} \|^2 \tag{3}$$

The refined bottleneck $h_K$ is then used for final task prediction.

## 4 EXPERIMENTS

**Dataset.** We conduct our experiments on rs-fMRI scans from the Alzheimer's Disease Neuroimaging Initiative (ADNI) repository, a large-scale, multi-site study providing longi-

Table 1: CDR degradation prediction results across three time horizons.

| Model | Metric | 1-Year Horizon | 2-Year Horizon | 3-Year Horizon |
|---|---|---|---|---|
| Logistic Regression | F1 | 0.561 | 0.660 | 0.650 |
| | Accuracy | 0.759 | 0.737 | 0.701 |
| Multi Task-NN | F1 | 0.182 | 0.300 | 0.494 |
| | Accuracy | 0.740 | 0.687 | 0.700 |
| Single Task-NN | F1 | 0.233 | 0.423 | 0.440 |
| | Accuracy | 0.760 | 0.700 | 0.664 |
| AG-VIT | F1 | 0.710 | 0.768 | 0.749 |
| | Accuracy | 0.867 | 0.816 | 0.792 |
| **AG-VIT + TTA** | **F1** | **0.713** | **0.796** | **0.766** |
| | **Accuracy** | **0.870** | **0.843** | **0.805** |

tudinal neuroimaging, clinical, and cognitive data. We included all subjects with available resting-state fMRI (rs-fMRI) scans. Our dataset includes subjects across the cognitive spectrum, from Cognitively Normal (CN) to Mild Cognitive Impairment (MCI) and Alzheimer's Disease (AD).

**Downstream Tasks.** Our primary clinical application is the prediction of cognitive deterioration. Each subject in ADNI undergoes multiple Clinical Dementia Rating (CDR) assessments over time, providing a longitudinal measure of cognitive and functional impairment. The CDR scale ranges from 0 (no dementia) to 3 (severe dementia), with intermediate values of 0.5 (questionable), 1 (mild), and 2 (moderate). We frame cognitive decline prediction as a binary classification task, where a patient's label is set to ′1′ if the CDR score increased from its baseline assessment in a given time frame. We repeated the experiment for three different timeframes: (i) One year, (ii) Two years, and (iii) Three years. For this task, subjects without follow-up CDR assessments in the respective time windows were excluded. This formulation allows us to evaluate the model's ability to detect subtle neurophysiological patterns predictive of cognitive decline at different temporal horizons. To further validate the generalizability of our learned representations, we evaluated our model on age and gender classification. The results for these tasks are given in Appendix E.

**Baselines.** In AG-VIT, we train a 2-layer neural network on the set of features obtained after the pretraining phase. We compare the outcome of AG-VIT to three classifiers trained on features that constitute the upper triangular elements of the functional connectivity matrix. Recall that $R$ denotes the number of regions obtained by an Atlas parcellation $A \in \mathbb{R}^{R \times T}$. The elements of the functional connectivity matrix, denoted $C(i,j)$, are equal to the Pearson correlation between the BOLD signal samples of regions $i$ and $j$ in $A$. Training a classifier on such features is a common practice in neuroimaging, see Finn et al. (2015); Du et al. (2018) and references therein.

The FC feature set serves as the input for three baseline models: (i) **Logistic Regression (LR):** A standard linear classifier that provides a simple, interpretable performance benchmark. (ii) **Single-Task Neural Network (ST-NN):** A 2-layer MLP, matching the architecture of our model's fine-tuning heads. A separate ST-NN model was trained independently for each downstream task. (iii) **Multi-Task Neural Network (MT-NN):** A 2-layer MLP with a shared hidden layer and separate output heads for each prediction task. This model, trained jointly on all tasks, is used to assess any benefits of multi-task learning on connectivity features.

## 4.1 RESULTS

**Cognitive Decline Prediction.**

The primary results for CDR degradation prediction are shown in Table 1. Our proposed model significantly outperforms all baselines baselines across all three time horizons. For the 1-year prediction task, our model achieves an F1-score of 0.710, **an improvement of 14.9**

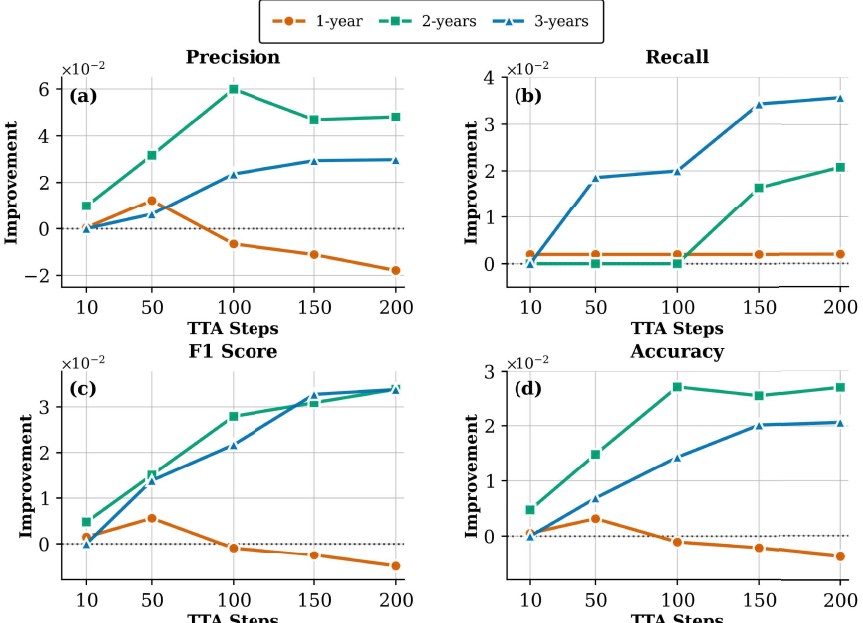

Figure 4: Test-time adaptation improvement for different metrics and prediction horizons. Each subplot shows the performance improvement (relative to no adaptation) as a function of TTA steps (10-200). Results are shown for 1-year (orange), 2-year (green), and 3-year (blue) CDR degradation prediction tasks.

**percentage points** over the best-performing baseline. This highlights the effectiveness of our atlas-based pretraining in learning features that are highly sensitive to subtle, early-stage neurophysiological changes predictive of decline. Crucially, the addition of our proposed Test-Time Adaptation (TTA) mechanism provides a further performance boost. For the 2-year and 3-year prediction horizons, TTA improves the F1-scores from 0.768 to 0.796 and from 0.749 to 0.766, respectively. This consistent improvement validates the benefit of personalizing the model at test time.

**Optimizing Test-Time Adaptation.** To determine the number of adaptation steps, we evaluated the TTA mechanism on a separate validation set. The results, presented in Fig. 4, show a slightly different behavior for the different tasks. Based on the validation set, we set $K = 100$ for the 2-year and 3-year horizons to capture the peak benefit shown in the longer-term prediction scenarios. For the more sensitive 1-year task, we used a more conservative $K = 50$ to leverage the initial gains without risking performance degradation from over-adaptation.

## 4.2 Model Explainability and Regional Importance Analysis

**Ablation-Based Importance Mapping.** To understand which brain regions contribute most critically to cognitive decline predictions, we developed a systematic ablation framework to quantify spatial patch importance. Unlike traditional gradient-based attribution methods, our approach directly measures the relevance of each spatial region by observing prediction degradation when that region's information is corrupted.

**Methodology.** We employed a mean ablation (Nanda et al., 2023) strategy to assess patch importance. For each spatial patch $\mathbf{P}_{t,n}$ in a test scan, we replaced it with the mean patch template computed across all training scans: $\bar{\mathbf{P}}_{t,n} = \frac{1}{N_{\text{train}}} \sum_{i=1}^{N_{\text{train}}} \mathbf{P}_{t,n}^{(i)}$. This approach replaces region-specific information with population-average patterns, effectively removing subject-specific functional signatures while maintaining realistic signal characteristics. For

each test scan and each spatial patch location $n$, we computed the importance score by,

$$\mathcal{I}_n = \mathcal{M}(\mathbf{X}) - \mathcal{M}(\mathbf{X}_n^{\text{ablated}}), \qquad (4)$$

where $\mathcal{M}$ represents a performance metric (F1, precision, recall, or accuracy), $\mathbf{X}$ is the original scan, and $\tilde{\mathbf{X}}_n$ is the scan with patch $n$ replaced by its mean template. A large positive value of $\mathcal{I}_n$ indicates that corrupting patch $n$ substantially degrades model performance, suggesting high importance for that spatial location. Note that the model parameters obtained in the two training stages remained fixed during this analysis, ensuring that the importance scores reflect the learned representations rather than any adaptation effects.

**Regional Aggregation.** To provide clinically interpretable results, we mapped the patch-level importance scores back to anatomical regions defined by the Schaefer-200 atlas. Each patch was assigned to its corresponding atlas region based on maximal spatial overlap. The regional importance score was computed as:

$$\mathcal{R}_r = \frac{1}{|\mathcal{N}_r|} \sum_{n \in \mathcal{N}_r} \mathcal{I}_n \qquad (5)$$

where $\mathcal{N}_r$ is the set of patches whose maximal overlap is with region $r$. This aggregation provides a brain-wide importance map at the resolution of functional brain networks, enabling direct clinical interpretation. The importance score was computed separately for the 1-year, 2-year, and 3-year CDR degradation predictions. For each task, we computed $\mathcal{R}_r$ for four performance metric: (i) F1 score (ii) Precision (iii) Recall (iv), and Accuracy.

**Comparing brain regions contributing to model prediction and $A\beta$ deposition.** To assess the relationship between rs-fMRI–derived predictions and established markers of Alzheimer's disease (AD) neuropathology, we compared the importance scores to the degree of amyloid $\beta$ ($A\beta$) accumulation across the brain (Haass & Selkoe, 2007). $A\beta$ is one of the two central molecular hallmarks of AD (alongside hyperphosphorylated tau) and is considered both the earliest to manifest and the more disease-specific hallmark of the two (Hardy & Selkoe, 2002) . Estimates of AD-related $A\beta$ deposition were derived from a cohort of 45 subjects with clinically established AD and 34 young healthy controls who underwent carbon-11–labeled Pittsburgh Compound B (PiB) PET imaging (Cohen & Klunk, 2014). PiB is injected intravenously, selectively binds to insoluble deposits of $A\beta$ in the brain, and scanned via well-established positron emission tomography (PET) (Driscoll et al., 2012). Here, we have used a cohort that forms the basis of the "Centiloid protocol", which is the only quantitative metric of $A\beta$ burden, recognized by the FDA (Rowe et al., 2016). While the Centiloid protocol is typically applied at the individual-subject level, here we alternatively applied a general linear model (GLM) contrasting AD and young control (YC) uptake. Preprocessing steps included PET-to-structural MRI coregistration, normalization to MNI space, within-subject PiB normalization to mean cerebellar uptake, and parcellation using the Schaefer200 atlas (Schaefer et al., 2018) (Fig. 5(C)). GLM results were Bonferroni-corrected for multiple comparisons.

To compare the importance scores for the 3-year prediction with $A\beta$ deposition in AD, we plotted Z-scores for (1) the 10 brain regions with the highest importance scores (blue) and (2) t-values derived from the AD > YC PiB contrast (Fig. 5(A)). Complementarily, to visually compare the patterns of brain regions contributing to model prediction and $A\beta$ deposition, we projected importance scores for the 3-year prediction (Fig. 5(B)), and the AD > YC PiB GLM contrast results (Fig. 5(A)) onto inflated 3D brain models.

**Results.** Comparing importance scores and the pattern of $A\beta$ uptake demonstrates partial concordance between regions with high importance scores and those with elevated $A\beta$ uptake. Specifically, several regions demonstrated both high importance and high $A\beta$ deposition, including the right Dorsal Attention Network, left Default Mode Network (temporal region), and right Default Mode Network (dorsal/medial prefrontal cortex) (Fig. 5(A)). Interestingly, multiple other regions exhibited a discordant pattern, with high importance scores but low $A\beta$ uptake. These included regions of the Visual, Somatomotor, and right Salience/Ventral Attention Networks (Fig. 5(A)).

**Discussion.** The partial concordance between importance scores and $A\beta$ uptake may be explained by several mechanisms. AD progression is widely considered to arise from a multitude of interacting factors. The most extensively studied—yet not fully understood—is

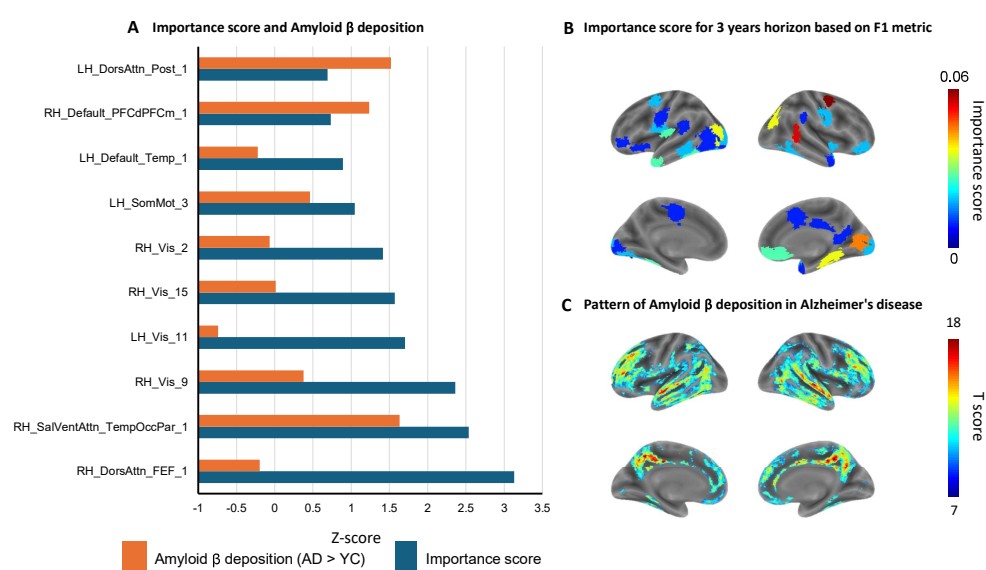

Figure 5: **Comparing brain regions contributing to model prediction and amyloid-β deposition**. (A) Z-scores for the 10 brain regions with the highest importance scores (blue) alongside t-values derived from the Alzheimer's vs. young control general linear model (GLM) contrasts in the matching brain regions (see abbreviations). (B). Importance scores for the 3-year prediction (based on the F1 metric) projected onto the corresponding regions of interest and displayed on an inflated 3D brain model. C. Results from the GLM analysis comparing PET PiB uptake between Alzheimer's patients and young control subjects, reflecting $A\beta$ deposition, projected onto an inflated 3D brain model. Brain regions abbreviations are in Table 4.

the interplay between $A\beta$ and tau (Nelson et al., 2012): while $A\beta$ is regarded as an early and disease-specific hallmark of AD, tau deposition has been suggested to spread along rs-fMRI–derived functional connectivity networks (Vogel et al., 2020). Accordingly, comparisons with tau deposition patterns may yield more concordant results. Given that rs-fMRI is primarily applied to analyze functional connectivity, signal alterations within specific regions may reflect disruptions within the network. Thus, one may reconsider $A\beta$ deposition as "concordant" when it occupies nodes functionally connected to regions with high importance scores. Finally, the model may accentuate previously unrecognized markers of regional vulnerability or, alternatively, highlight regions that are highly stable and relatively unaffected—similarly to the role the cerebellum plays in within-subject PiB normalization (Heeman et al., 2020).

**Limitations.** In our work, we demonstrated that transformer-based representations of rs-fMRI generated by AG-ViT achieve significant performance gains over conventional approaches and open new avenues for interpreting the functional substrates of Alzheimer's disease. However, several limitations should be considered in future work: (i) AG-ViT is trained on approximately 900 scans from the ADNI repository, most of which are from patients in North America. Incorporating additional datasets - including non-Alzheimer's cohorts - may improve the model's generalization across populations and clinical contexts. (ii) fMRI scans primarily capture functional connectivity patterns. Extending the architecture to integrate additional imaging modalities, such as PET, could provide complementary information and further enhance predictive performance. (iii) The brain regions identified as having the strongest impact on model predictions show only partial concordance with patterns of $A\beta$ uptake. Incorporating comparisons with tau deposition, or other pathological markers, may improve the interpretability and biological relevance of the model.

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
