# OpenReview forum: "AG-ViT: Atlas-Guided Vision Transformer for Predicting Cognitive Decline"
_ICLR.cc/2026/Conference — ICLR 2026 Conference Desk Rejected Submission_

### Official Review · Reviewer_qBDF · 2025-10-26

**Soundness:** 1
**Presentation:** 1
**Contribution:** 1
**Rating:** 2
**Confidence:** 4

**Summary:**

This paper proposes a attention-based pretraining scheme to reduce a high dimensionality of resting state fMRI data. The model yields low dimensional embedding features, and these features are used to estimate the probability of cognitive decline. Experiments were performed on the ADNI dataset. Neuroscientific analyses, e.g., brain regional analyses, were performed to identify significant brain regions that affected model predictions.

**Strengths:**

1)	The paper includes statistical analysis to enhance model interpretability, which is important to understand medical data. Also, multiple metrics were used to evaluate the model performance.

**Weaknesses:**

1)	Lack of technical novelty. The proposed AG-ViT does not introduce a fundamentally new learning mechanism. The model is simply built upon the existing ViT architecture with fine-tuning of additional bottleneck features, offering only incremental modifications. Specifically, the benefit of the pretraining step incorporating atlas representation is unclear. Compared to the typical ViT, what is the concrete advantage of learning bottleneck features through the atlas decoder, and how does this component meaningfully contribute to model performance or interpretability?

2)	Given that there are many existing studies for feature processing, this paper does not explain any motivations and rationales to specifically choose the attention mechanism for solving the task.

3)	Significant lack of baseline models and insufficient data usage. Basically, only two basic ML frameworks (i.e., logistic regression and 2-layer MLP) were used, without adopting recent models. More comparisons with recent studies and additional datasets are necessary to sufficiently evaluate the generalizability of the proposed model. For example, [1,2,3,4,5] can be adopted for additional baselines. Moreover, given that 4D (brain) image classification and brain network-based studies are heavily researched topics, I believe that authors can easily find more studies on this task with open source codes.

[1] Kan et al., “Brain network transformer”, NeurIPS 2022.

[2] Kawahara et al. "BrainNetCNN: Convolutional neural networks for brain networks; towards predicting neurodevelopment." NeuroImage 2017.

[3] Li et al. "Braingnn: Interpretable brain graph neural network for fmri analysis." Medical Image Analysis, 2021.

[4] Jiang et al., “Characterizing functional brain networks via Spatio-Temporal Attention 4D Convolutional Neural Networks”, Neural Networks, 2023.

[5] Choy et al., “4D Spatio-Temporal ConvNets: Minkowski Convolutional Neural Networks”, CVPR, 2019.

4)	The identified brain regions with high important scores do not generally align with the patterns of A$\beta$ uptake, which raises questions about the biological plausibility of the model’s decision process.

**Questions:**

1)	What is the main technical novelty compared to recent studies?

2)	Why should the attention mechanism be used, and what is the advantage of it for solving the task compared to other studies?

---

### Official Review · Reviewer_UeV1 · 2025-10-30

**Soundness:** 2
**Presentation:** 2
**Contribution:** 2
**Rating:** 0
**Confidence:** 5

**Summary:**

This paper proposed an atlas-guided VIT for AD prediction. The proposed method consists of a self-supervised pretraining method and a supervised finetuning stage. The proposed method is validated on ADNI datasets. The experimental results show that the proposed method  achieve a better performance when compared with the competing methods.

**Strengths:**

1. An atlas-based self-supervised pretraining method is proposed in this work.
2. The experimental results are better than the competing methods.

**Weaknesses:**

This study has several methodological limitations that warrant attention. The reliance solely on the Schaefer 2018 parcellation method raises concerns about the robustness of findings across different brain atlases. While proposing an atlas-based self-supervised learning approach, the authors fail to establish its advantages over existing self-supervised methods. The experimental design is constrained by simplistic baseline comparisons lacking state-of-the-art benchmarks, limited evaluation metrics (only F1-score and Accuracy), and insufficient methodological details regarding image preprocessing and data partitioning. Furthermore, the exclusive validation on the ADNI dataset without external testing undermines claims about the method's generalization capability. These issues collectively impact the reliability and generalizability of the reported results.

**Questions:**

1. This paper employed the Schaefer 2018 parcellation method to obtain the brain atlas. In reality, there are numerous atlas parcellation methods. Whether different parcellation methods would significantly impact the methodology was not mentioned in the paper.
2. An important innovation of this paper is its atlas-based self-supervised learning method. There are many self-supervised learning methods proposed; the authors did not discuss the advantages of their proposed method compared to these methods.
3. The comparative methods used in the paper are overly simplistic and lack comparison with state-of-the-art methods.
4. The evaluation metrics are limited to only F1-score and Accuracy.
5. The experimental description lacks crucial details, such as image preprocessing procedures and dataset partitioning.
6. The experiments were only validated on the ADNI dataset, lacking external dataset validation to demonstrate the method's generalization capability.

---

### Official Review · Reviewer_U82q · 2025-10-31

**Soundness:** 2
**Presentation:** 1
**Contribution:** 2
**Rating:** 2
**Confidence:** 4

**Summary:**

This study presents an approach for predicting the probability of cognitive decline in Alzheimer's disease patients by developing AG-ViT, a model that learns low-dimensional embeddings from 4D rs-fMRI scan data. As its core methodology, the model employs a self-supervised pretraining strategy, training it to reconstruct 2D atlas-based temporal representations from the original 4D fMRI scans. The embeddings learned through this process are then used as input for a downstream network that predicts cognitive decline.
Trained on rs-fMRI scan data from the ADNI, the model demonstrated improved performance compared to existing functional connectivity-based models. Furthermore, performance was additionally enhanced using a TTA technique, and the study also demonstrated an interpretability methodology to identify brain regions that significantly impact the model's predictions.

**Strengths:**

The motivation to incorporate neuroanatomical priors (via atlas guidance) into transformer architectures for rs-fMRI analysis is reasonable and timely.
The integration of test-time adaptation for clinical stability adds a practical dimension, aligning with deployment concerns in medical AI.

**Weaknesses:**

Except for the atlas-guided component, AG-ViT represents only an incremental improvement over prior works. Although the authors hypothesize that reconstructing functionally meaningful regional dynamics through atlas parcellation enhances robustness relative to voxel-wise or masked regional representations, this claim is not substantiated by quantitative evidence or dedicated ablation experiments in the manuscript.
The manuscript frequently claims that pairwise FC-based methods are suboptimal but provides limited experimental or theoretical justification. While FC-based baselines are included, comparisons are superficial (F1/Accuracy only), lacking interpretability or clinical analyses that demonstrate genuine superiority.

The proposed model relies on a specific Schaefer atlas, but robustness across different parcellations (Craddock, Power, etc.) is not tested.

The paper omits detailed descriptions of dataset splits (train/val/test ratios, subject-level independence). This omission hinders reproducibility and fair evaluation.

Results are presented as single scores without variance or significance testing, undermining the statistical reliability of claims.

Comparative models are limited to logistic regression and simple neural networks. The absence of modern baselines, such as Brain Transformer, BrainLM, SwiFT, or self-supervised/MAE-based approaches, reduces the credibility of reported performance gains.

Some notations and arrows in Figures 2 and 3 are too small and the colors lack sufficient contrast, making them difficult to read.

Additionally, certain parts of Figure 2 do not appear to be consistent (e.g., Figure 2b) with the corresponding descriptions in the main text.

**Questions:**

The model uses a fixed 3D patch size of 6×6×6. What motivated this choice? Was it empirically validated or based on prior literature? A sensitivity analysis could clarify model stability.

Although the abstract and introduction describe the task as a probabilistic prediction of cognitive decline, the experimental design reformulates it as a binary classification problem, labeling ‘1’ when a subject’s CDR score increases within a specified time window and ‘0’ otherwise. Furthermore, the evaluation relies solely on deterministic metrics such as F1 score and accuracy, without employing any calibration- or probability-based measures. This discrepancy between the stated objective and the implemented methodology detracts from the conceptual clarity of the manuscript and raises concerns regarding the validity of framing the problem as a probabilistic prediction.

The authors emphasize the efficiency benefit of using an atlas-guided decoder with fewer parameters during training. However, the final model also incorporates a TTA procedure, which involves additional gradient-based update steps and thus introduces extra computational cost. This raises a question: in practical deployment scenarios, wouldn’t a model that incurs higher cost during training but operates faster and more efficiently at inference be more desirable than the proposed structure?

---

### Official Review · Reviewer_xSyh · 2025-11-01

**Soundness:** 3
**Presentation:** 2
**Contribution:** 2
**Rating:** 4
**Confidence:** 3

**Summary:**

This paper presents an atlas-guided vision transformer (AG-ViT) or longitudinal prediction in Alzheimer’s disease. The model is trained to predict clinical status at a future time horizon (e.g., 1-3 years), and it produces region-level importance scores. These scores are then compared against Alzheimer’s disease pathology patterns, in order to support biological interpretability.

**Strengths:**

1. The introduction of an explicit prediction time horizon (e.g., 3-year outcome) is clinically meaningful.

2. The authors compare the most important regions highlighted by the model with established biological markers of Alzheimer’s disease pathology, which makes the interpretation concrete and clinically relevant.

**Weaknesses:**

1.  When reporting gains in metrics over standard baselines, please include significance tests or confidence intervals. This is especially important given the limited dataset size.

2. ADNI (and similar longitudinal Alzheimer’s cohorts) is not extremely large. A transformer-style model can be high capacity. The paper should report model size (number of parameters), training cost, and any regularization strategies used. How do you mitigate overfitting? This matters for generalizability.

3. The method is evaluated for a 3-year horizon. Have you also evaluated longer horizons?

**Questions:**

The authors propose an atlas-guided ViT instead of a voxel-wise approach. Can you explain why atlas guidance is especially suitable for Alzheimer’s disease? And how sensitive are the results to the choice of atlas? If a different atlas (different resolution, different network definitions) is used, do you obtain similar importance maps and predictions?

---

### Author Response · Authors · 2025-12-03

We thank the reviewers for the time and effort invested in evaluating our work. We also thank the reviewers for their comments and suggestions. Here, we address the reviewers' main concerns.

1. **Overall assessment of contribution**: We made two main contributions in our paper:

    * *Methodological contribution to the field for analyzing fMRI scans using attention mechanisms*. Our paper introduces an Atlas-Guided Transformer architecture for analyzing resting-state fMRI scans. In most fMRI analysis applications, the number of available scans with relevant clinical data is limited. A general-purpose repository such as the UK Biobank may contain 70,000 scans, but to address clinical questions of significance for specific diseases such as Alzheimer's disease (AD), PTSD, and depression, there is a need for customized datasets that include additional modalities (i.e. specific PET scans for Alzheimer's), as well as relevant clinical assessments. The number of samples in such repositories is limited. ADNI, the most relevant repository for Alzheimer's research, includes only ~1000 patients. Training a transformer on such small datasets is challenging.

      Our proposed architecture reduces the number of parameters substantially while imposing biologically meaningful structure derived from cortical and subcortical atlases. This makes the training process much more effective. We tried to implement several competing transformer-based algorithms, such as SWIFT, but the network overfitted, and the prediction results were much worse than what we obtained when Atlas was used for output as a preliminary pre-training signal.


    * *Contribution to the specific Field of Alzheimer's disease*. We address a clinically important and non-trivial question: predicting short-term cognitive decline (1–3 years) based solely on baseline rs-fMRI scans. The ability to obtain accurate predictions for this time frame based on rs-fMRI scans is remarkable. Given recent approvals of new medicines, such a prediction may be relevant to clinical decisions, such as prescribing a specific medicine with serious side effects. In addition, we detect brain regions whose connectivity patterns are significant to the decline, and compare them to Amyloid beta deposition.


2. **Comparison to relevant methods**. Several reviewers expressed concerns about comparisons with existing attention-based neuroimaging methods. We made substantial efforts to reproduce and evaluate state-of-the-art approaches, including tff, BrainLM, and SWIFT. Unfortunately, tff and BrainLM could not be successfully adapted to ADNI despite significant effort; the publicly available codebases lack crucial documentation (e.g., data pre-processing specifications, input layouts, hyperparameter details), making generalization to new datasets extremely difficult.
We were able to apply SWIFT to the ADNI data; however, the model overfitted heavily, and its predictive performance was markedly below the baselines included in our submission. These observations reinforce our motivation for designing the Atlas-Guided architecture specifically for small-sample clinical neuroimaging settings.

3. **Stability to choice of Atlas and patch size** Several reviewers requested additional experiments to test the robustness of our approach. We conducted an extensive atlas baseline test to determine the most effective brain parcellation for our task. The evaluation included a range of atlases: ["schaefer100", 'schaefer200', "schaefer400", "schaefer600", "power2011", "oxford-harvard", "destrieux"]. For each parcellation, the Functional Connectivity (FC) matrix was extracted and utilized as features in two basic models (Linear Regression and MLP). The Schaefer atlases with 400 regions and 600 regions usually outperformed the others. For example, applying LR to predict sex yielded an F1 score of 0.89 with 600 regions, and 0.85 with 200. For age prediction, We intend to further explore the effect of different parcellations on the results, as this indeed might have a significant effect on the results.

   Choice of patch size – We varied layers (4, 8, 12), heads (4, 8), and patches (6, 8, 12). Loss ranged from 0.277–0.543. Top results were consistent, confirming robustness; 4 layers with 6×6×6 patches performed best. Performance dropped only with deeper models (12 layers) or or very large patches(12x12x12), likely due to overfitting on limited data and loss of local information, respectively. We added the results to the supplementary material.

4. **Specification training details** We added details such as the number of parameters and computational cost of our model and train/validation/testing split.

---

### Note · Program_Chairs · 2026-01-17
**Submission Desk Rejected by Program Chairs**

The following references in this submission do not refer to real documents and/or have major errors in bibliographic information:

     Sarah Parisot, Sofia I Ktena, Enzo Ferrante, Martin Lee, Ricardo Guerrero, Ben Glocker, and Daniel Rueckert. A graph convolutional neural network for classification of mild cognitive impairment. Brain, 142(4):1032-1045, 2019.